# Long-term comparison between index selection and optimal independent culling in plant breeding programs with genomic prediction

**Lorena G. Batista**[1]*, **Robert Chris Gaynor**[2], **Gabriel R. A. Margarido**[1], **Tim Byrne**[3],
**Peter Amer**[4], **Gregor Gorjanc**[2], **John M. Hickey**[2]

**1** "Luiz de Queiroz" College of Agriculture (University of São Paulo), Piracicaba, SP, Brazil, **2** The Roslin
Institute and Royal (Dick) School of Veterinary Studies, University of Edinburgh, Easter Bush Campus
Research Centre, Midlothian, United Kingdom, **3** Abacusbio, Roslin Innovation Centre, University of
Edinburgh, Easter Bush Campus Research Centre, Midlothian, United Kingdom, **4** Abacusbio, Dunedin,
New Zealand

* loregbatista@gmail.com

pone.0235554

UNITED STATES

**Data Availability Statement:** All relevant data are
within the manuscript and its Supporting
Information files.

## Abstract

In the context of genomic selection, we evaluated and compared breeding programs using
either index selection or independent culling for recurrent selection of parents. We simulated
a clonally propagated crop breeding program for 20 cycles using either independent culling
or an economic index with two unfavourably correlated traits under selection. Cycle time
from crossing to selection of parents was kept the same for both strategies. Both methods
led to increasingly unfavourable genetic correlations between traits and, compared to inde-
pendent culling, index selection led to larger changes in the genetic correlation between the
two traits. When linkage disequilibrium was not considered, the two methods had similar
losses of genetic diversity. Two independent culling approaches were evaluated, one using
optimal culling levels and one using the same selection intensity for both traits. Optimal cull-
ing levels outperformed the same selection intensity even when traits had the same eco-
nomic importance. Therefore, accurately estimating optimal culling levels is essential for
maximizing gains when independent culling is performed. Once optimal culling levels are
achieved, independent culling and index selection lead to comparable genetic gains.

## Introduction

Crop breeding seeks to develop improved cultivars. Besides high yield levels, a successful culti-
var in many crops must meet minimal standards for several other traits that are economically
important, such as pest and disease resistance and product quality. Traits are often unfavour-
ably correlated with each other [e.g., 1–5]. When traits are antagonistically correlated, selection
for one trait causes an undesired economic response in the other trait [6, 7]. This makes breed-
ing to simultaneously improve multiple traits complicated.

**Funding:** LGB was supported by the Coordenação de Aperfeiçoamento de Pessoal de Nível Superior (CAPES, Computational Biology Programme, Grant No. BEX 0043/17-6, Finance Code 001). JMH, RCG and GG acknowledge the financial support from BBSRC and KWS UK, RAGT Seeds Ltd., Elsoms Wheat Ltd and Limagrain UK for the project "GplusE: Genomic selection and Environment modelling for next generation wheat breeding" (grants BB/L022141/1 and BB/L020467/1). The funders had no role in study design, data collection and analysis, decision to publish, or preparation of the manuscript. The funder Abacusbio provided support in the form of salaries for authors TB and PA but did not have any additional role in the study design, data collection and analysis, decision to publish, or preparation of the manuscript. The specific roles of these authors are articulated in the 'author contributions' section.

**Competing interests:** The commercial affiliation with Abacusbio does not alter our adherence to PLOS ONE policies on sharing data and materials

Independent culling and the use of a selection index are two commonly used methods in plant breeding programs for selecting for multiple traits [7]. Independent culling involves establishing minimum standards (i.e., culling levels) for each trait and selecting only individuals that meet these minimum standards. The thresholds can be set according to a specific selection intensity or a specific value, such as a value relative to an agronomic check. The application of independent culling can be to multiple traits simultaneously or to individual traits sequentially. The selection index method involves selection for all traits simultaneously based on a linear or nonlinear combination of individual traits weighted by their importance for the breeding objective [8].

Theoretically, the selection index is the most effective method of selection for multiple traits [8–10]. Independent culling is less effective than index selection because, when strictly applied, it will not select individuals below the threshold for only one trait despite being exceptional for all other traits, while the use of a selection index makes it possible to retain those individuals [7]. However, independent culling can achieve nearly equal effectiveness using optimised thresholds [11].

When cost is considered, independent culling can be more efficient than a selection index [11]. This is because independent culling does not require phenotypes for all traits at one time, whereas strict application of a selection index requires phenotypes for all traits. This benefit is particularly valuable to plant breeders, because early stages of the breeding program often have a very large number of individuals. Phenotyping all individuals for all traits is likely to be logistically and financially infeasible. For example, some traits have a high measurement cost, such as bread quality in wheat, so that they cannot be measured on a large number of individuals. Furthermore, some traits can be measured only on older plants, such as lifetime production in sugarcane, or on a plot or group basis. Delaying selection until these traits become available would be effectively equal to random selection, because the breeder would have to reduce the overall size of the early stage. Thus, practical constraints require at least some use of independent culling on traits that can be phenotyped simply/quickly and at a lower cost in breeding programs utilising phenotypic selection.

Genomic selection in plant breeding may render the cost efficiency benefit of independent culling irrelevant if all early generation individuals are genotyped. This is because genomic selection allows accurate prediction of all traits at once [12]. While genotyping all early generation individuals is not standard in most current breeding programs, it may become so in the future. This is likely to be the case if breeding programs adopt a two-part strategy to breeding that explicitly splits breeding programs into a rapid cycling, genomic selection guided, population improvement part tasked with developing new germplasm and a product development part focused on developing new varieties. Simulations of these breeding programs suggest they can deliver considerably more genetic gain than more conventional breeding programs [13].

Another reason why independent culling is often preferred over index selection is the need to correctly define the economic model for selection indices to be successful. However, what breeders often overlook is the fact that the accurate estimation of economic weights is required to maximize gains even under independent culling. This is because there is a combination of selection intensities for the traits that maximizes the genetic gain (optimal culling levels) and economic weights need to be accounted for when estimating optimal culling levels [11, 14, 15]. Therefore, regardless of the method of selection, plant breeders would benefit greatly from an increased emphasis on understanding and quantifying the economics of their species and using more analytical approaches when selecting for multiple traits.

Several studies have already shown the benefits of incorporating genomic selection strategies into crop breeding programs [13, 16–18]. Other studies have demonstrated that combining index selection and genomic prediction can increase genetic gain in breeding programs [19, 20] and, in the long term, even higher genetic gains can be obtained when multi-trait

optimization strategies that also control for the loss of genetic diversity are used [21]. However, differences in how multi-trait selection methods can affect not only genetic gain but other population parameters such as genetic diversity and genetic correlations over several cycles of recurrent selection have not yet been thoroughly investigated. In order to provide a more detailed account of population dynamics in the genomic selection framework, we used simulations of recurrent breeding programs to evaluate and compare both index selection and the independent culling method for 20 cycles of selection. The purpose of these simulations was to quantify the difference between optimally set independent culling levels and an optimal selection index. The simulations also investigated the sensitivity of independent culling using a sub-optimal culling level.

## Material and methods

Stochastic simulations of entire breeding programs for multiple traits were used to compare the genetic gains in a breeding program using independent culling levels and a breeding program using an economic selection index for selection of parents. In the independent culling approach, selection was performed for one trait at a time at each stage of selection. A clonally propagated crop species was considered. Generally, in breeding programs for clonally propagated species, several crosses are performed between highly heterozygous hybrids, and all the genotypes in the resulting $F_1$ progenies are candidate clones to be released as cultivars or used as parents in the next breeding cycle [22]. The methods were compared using the average of fifty replicates, each replicate consisting of: i) a burn-in phase shared by both strategies so that each strategy had an identical, realistic starting point; and ii) an evaluation phase that simulated future breeding with different breeding strategies. The burn-in phase consisted of 20 years of breeding using independent culling for the selection of parents and the evaluation phase consisted of 20 cycles of selection using either independent culling or index selection.

### Genome sequence

For each replicate, a genome consisting of 10 chromosome pairs was simulated for the hypothetical plant species. In order to assign realistic values of simulation parameters for a crop species, we chose AlphaSimR [23] default values for wheat. The chromosomes were assigned a genetic length of 1.43 Morgans and a physical length of $8x10^8$ base pairs. Sequences for each chromosome were generated using the Markovian Coalescent Simulator [24] and AlphaSimR. Recombination rate was inferred from genome size (i.e. 1.43 Morgans / $8x10^8$ base pairs = $1.8x10^{-9}$ per base pair), and mutation rate was set to $2x10^{-9}$ per base pair. Effective population size was set to 50, with linear piecewise increases to 1,000 at 100 generations ago, 6,000 at 1,000 generations ago, 12,000 at 10,000 generations ago, and 32,000 at 100,000 generations ago [25].

### Founder genotypes

Simulated genome sequences were used to produce 50 founder genotypes. These founder genotypes served as the initial parents in the burn-in phase. This was accomplished by randomly sampling gametes from the simulated genome to assign as sequences for the founders. Sites that were segregating in the founders' sequences were randomly selected to serve as 1,000 causal loci per chromosome (10,000 across the genome in total). To simulate genetic correlations between traits, the traits were treated as pleiotropic and the additive effects of the causal

loci alleles were sampled from a multivariate normal distribution with mean $\mu = \begin{bmatrix} 0 \\ 0 \end{bmatrix}$ and

desired values of correlation.

## Estimated breeding values

The true genetic value of each simulated trait was determined by the summing of its causal loci allele effects. The matrix $\mathbf{E}$ with the estimated breeding values of the traits for each individual in the population was obtained according to the formula:

$$\mathbf{E} = \mathbf{YP}^{-1}\mathbf{G}$$

where $\mathbf{Y}$ is the matrix of phenotypes simulated by adding random error to the true genetic values of the traits, where rows correspond to individuals in the population and columns correspond to traits. The random error was sampled from a multivariate normal distribution with mean $\mu = \begin{bmatrix} 0 \\ 0 \end{bmatrix}$ and zero covariance, with variance values tuned to achieve a target level of accuracy ($r$). In this study we define accuracy as the correlation between true and estimated breeding values. $\mathbf{P}$ is the phenotypic variance-covariance matrix of the traits, and $\mathbf{G}$ is the genetic variance-covariance matrix for the traits.

## Breeding methods

The simulations modelled breeding for two component traits (T1 and T2) that were improved using either independent culling or an economic selection index. With both strategies, an $F_1$ population of 5,000 individuals was generated in each cycle by randomly crossing the individuals in the crossing block (parents). With independent culling, selection was applied in two stages: a proportion of individuals was selected first based on T1 and then, from this proportion, the parents of the next breeding cycle were selected based on T2. With the selection index approach, the $F_1$ individuals with the highest values for the index trait were selected as parents of the next breeding cycle. The index trait ($\mathbf{I}$) was the sum of the estimated breeding values for each trait weighted by their economic importance:

$$\mathbf{I} = \mathbf{Ew}$$

where $\mathbf{E}$ is, as previously described, the matrix of estimated breeding values and $\mathbf{w}$ is the column vector of economic weights of the traits.

The number of selected parents (50 parents) and the cycle time from crossing to selection of new parents was kept the same for both strategies, so the comparisons between them reflect only differences due to the method of selection. The overall selection scheme used for each method of selection is shown in S1 Fig in S1 File. For simulation of breeding programs, we used the R package AlphaSimR. All codes used for the simulations are shown in S2 File.

## Simulated scenarios

The selection index and independent culling methods were compared in a set of scenarios that aimed to assess the relative performance of the methods under different levels of accuracy of selection, and relative economic importance of T2. We were interested in investigating only the relative performance of selection methods under challenging conditions for multi-trait selection. For this reason, only an unfavourable genetic correlation between traits was simulated. We used a value of -0.50 for the genetic correlation. A summary of all simulated scenarios we used in this study is shown in Table 1.

For one set of scenarios we simulated four levels of accuracy (0.3, 0.5, 0.7, and 0.99) and assigned the same economic importance for both traits. In another set of scenarios, we varied the relative economic importance of T2, but fixed selection accuracy at 0.7. Three levels of relative economic importance were simulated. T1 was given an economic importance of 1.0 and

**Table 1. Summary of parameters simulated in all comparison scenarios of recurrent selection breeding programs using either independent culling or selection index with two traits.**

| Scenario | Selected Proportion | | Relative economic importance of Trait 2 | Accuracy |
|---|---|---|---|---|
| | Trait 1 | Trait 2 | | |
| 1 | Optimum | Optimum | 1.0 | 0.3 |
| 2 | Optimum | Optimum | 1.0 | 0.5 |
| 3 | Optimum | Optimum | 1.0 | 0.99 |
| 4 | Optimum | Optimum | 1.0 | 0.7 |
| 5 | Optimum | Optimum | 2.5 | 0.7 |
| 6 | Optimum | Optimum | 5.0 | 0.7 |
| 7 | 10% | 10% | 1.0 | 0.7 |
| 8 | 10% | 10% | 2.5 | 0.7 |
| 9 | 10% | 10% | 5.0 | 0.7 |

T2 an economic importance of either 1.0, 2.5 or 5.0. For each level of relative economic importance, we simulated: i) scenarios where the proportion selected was the same (10%) for both traits, and ii) scenarios where the proportions selected were set to achieve optimal culling levels (i.e., optimal independent culling). To find the optimal proportions at each cycle, we fixed the number of parents selected (50 parents) and found the number of individuals to be selected in the first culling stage that maximized parents' economic value (measured as the index trait)., which was obtained based on the estimated breeding values. Thus, over the cycles of selection, when using optimal culling levels, instead of a fixed proportion selected of 10%, the proportion selected for each trait varied between cycles.

## Comparison

The comparisons were made in terms of: i) genetic gain ii) genetic diversity, iii) the efficiency of converting genetic diversity into genetic gain for the index; and iv) genetic correlation between traits. For genetic gain and genetic diversity, we report values based on the individuals in the crossing block (parents) at each cycle of selection. We measured genetic gain as the increment in genetic mean (average of true genetic values) compared to the genetic mean in year 20. We measured genetic diversity with genetic standard deviation and genic standard deviation. We calculated genetic standard deviation as standard deviation of true genetic values. We calculated genic standard deviation as $\sigma_a = \sqrt{2 \sum_{i=1}^{n_q} p_i(1 - p_i)\alpha_i^2}$, where $n_q$ is the number of causal loci and $p_i$ and $\alpha_i$ are, respectively, allele frequency and allele substitution effect at the $i$-th causal locus.

To measure efficiency, genetic mean and genic standard deviation were standardized to mean zero and unit standard deviation in year 20. We measured efficiency of converting genetic diversity into genetic gain by regressing the achieved genetic mean ($y_t = (\mu_{a_t} - \mu_{a_{20}})/\sigma_{a_{20}}^2$) on lost genetic diversity ($x_t = 1 - \sigma_{a_t}/\sigma_{a_{20}}$), i.e., $y_t = \alpha + bx_t + e_t$, where $b$ is efficiency [26]. We estimated efficiency with robust regression using function rlm() in R [27].

For genetic correlation, we report the correlation between the true genetic values of T1 and T2. We calculated this metric on the individuals in the $F_1$ population at each cycle of selection.

## Results

Index selection provided consistent genetic gains and was equivalent to independent culling in terms of genetic gains and efficiency when optimal culling levels were used. Index selection

performed better than independent culling in scenarios where independent culling was performed using the same selection intensity for each trait.

We have structured the description of the results in two parts, corresponding to how the relative performance of the selection methods was affected by: i) the accuracy of selection, and ii) the relative economic importance of traits.

## Accuracy of selection

Increases in accuracy accentuated the differences in the genotypes being selected by either independent culling or index selection. This is shown in Fig 1, where the genotypes selected as parents by each selection method are highlighted. Lower levels of accuracy led to a more diffuse cluster of selected genotypes and, with increasing selection accuracy, the cluster of selected genotypes approached what was expected for each method of selection [7].

Fig 2 shows the change in the genetic correlation between the component traits for both independent culling and index selection over 20 cycles of selection at different levels of accuracy. Both selection methods resulted in the correlation between traits becoming increasingly unfavourable over the cycles of selection. For both methods, the change in the genetic correlation increased with higher values of accuracy. Compared to independent culling, index selection led to larger changes in the genetic correlation between the two traits. After 20 cycles of selection with accuracy of 0.3, independent culling led to a genetic correlation that was 9% more unfavourable than the genetic correlation in cycle 0, while index selection led to a genetic correlation that was 17% more unfavourable than the genetic correlation in cycle 0. After 20 cycles of selection with accuracy of 0.99, independent culling led to a genetic correlation that

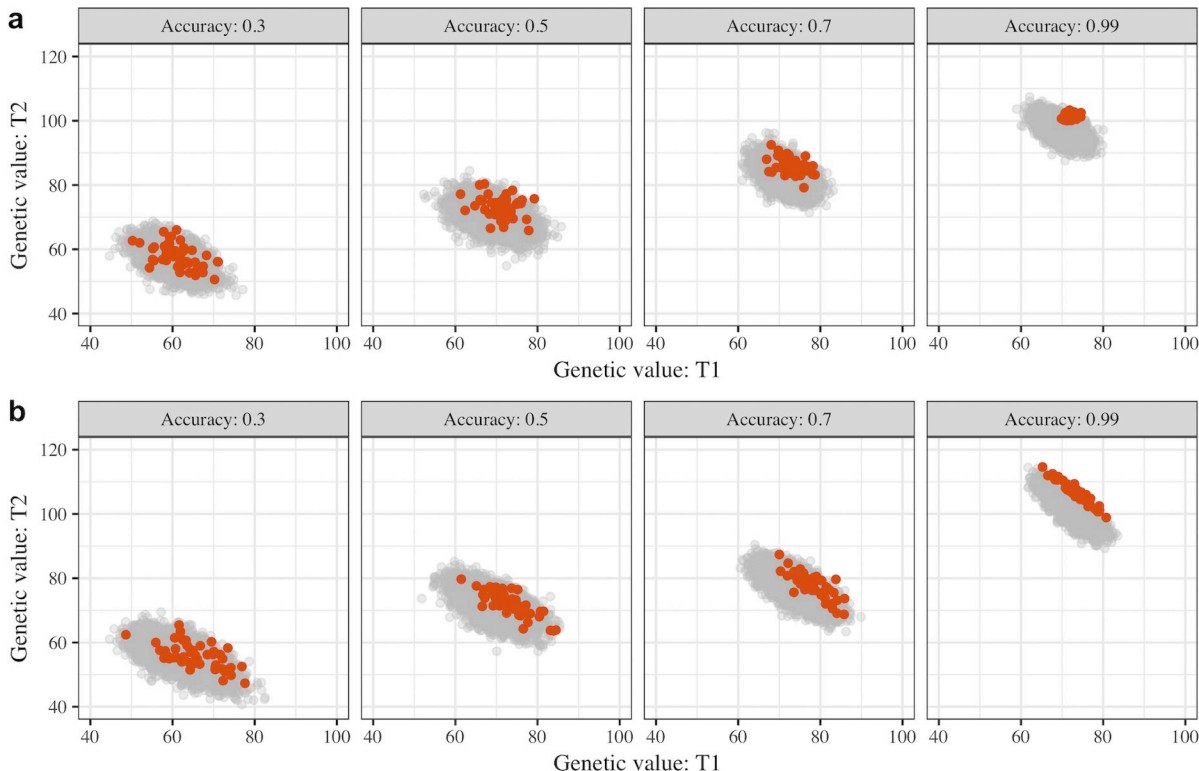

**Fig 1.** Scatterplots of true genetic values for Trait 1 (T1) and Trait 2 (T2) of the genotypes in the F1 population (grey) and genotypes selected as parents (orange) in the third cycle of selection using either independent culling (a) or a selection index (b) with different levels of accuracy.

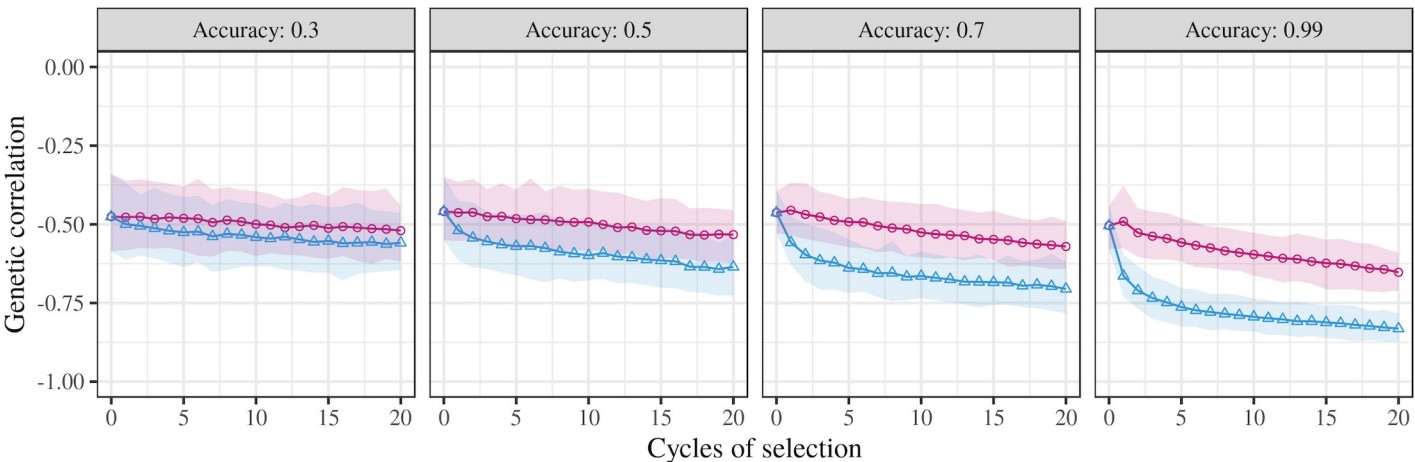

**Fig 2. Change in genetic correlation (mean and 95% confidence interval) between traits in the F1 population over 20 cycles of selection using either optimal independent culling (IC) or a selection index (SI) with different levels of accuracy, and Trait 2 relative economic importance of 1.0.**

was 29% more unfavourable than the genetic correlation in cycle 0, while index selection led to a genetic correlation that was 64% more unfavourable than the genetic correlation in cycle 0.

The change of genetic mean in parents for the component traits and the index trait over the cycles of selection using each method is shown in Fig 3. For both methods, the genetic gains for the component traits and the index trait increased with higher values of accuracy. In general, the selection index method and independent culling with optimal culling levels led to equivalent genetic gains for the component traits and the index trait. Only in the scenario with 0.99 accuracy did index selection lead to a slightly higher genetic gain than that achieved with optimal independent culling. For the index trait, after 20 cycles of selection with accuracy of 0.99, index selection had a genetic gain 4% higher than the genetic gain achieved with independent culling.

Table 2 shows the genetic standard deviation of parents in cycle 20 and the loss in genetic standard deviation in cycle 20 compared to the genetic standard deviation in cycle 0 for the component traits and the index trait. The change of genetic diversity in parents for the component traits and the index trait over the cycles of selection using each method is shown in S2 Fig in S1 File. For the component traits, under index selection, the genetic standard deviation showed an initial increase in the first few cycles of selection followed by a gradual decrease in the subsequent cycles. Under independent culling, the decrease in the genetic standard deviation of the component traits was continual over the cycles of selection. Both of these trends were more obvious with increasing values of accuracy. For all values of accuracy, independent culling led to a higher loss in the genetic standard deviation of the component traits compared to the index selection. For T1 and T2, independent culling with accuracy of 0.3 led to losses of genetic standard deviation that were respectively 6% and 5% higher than the loss of genetic standard deviation observed for index selection. With accuracy of 0.99, for T1 and T2 independent culling led to losses of genetic standard deviation that were respectively 65% and 51% higher than the losses of genetic standard deviation observed for index selection. For the index trait, both methods led to equivalent values of genetic standard deviation. With accuracies of 0.3 and 0.99, index selection led to a loss in the genetic standard deviation of the index trait that was 3% higher than the loss of genetic standard deviation observed using independent culling.

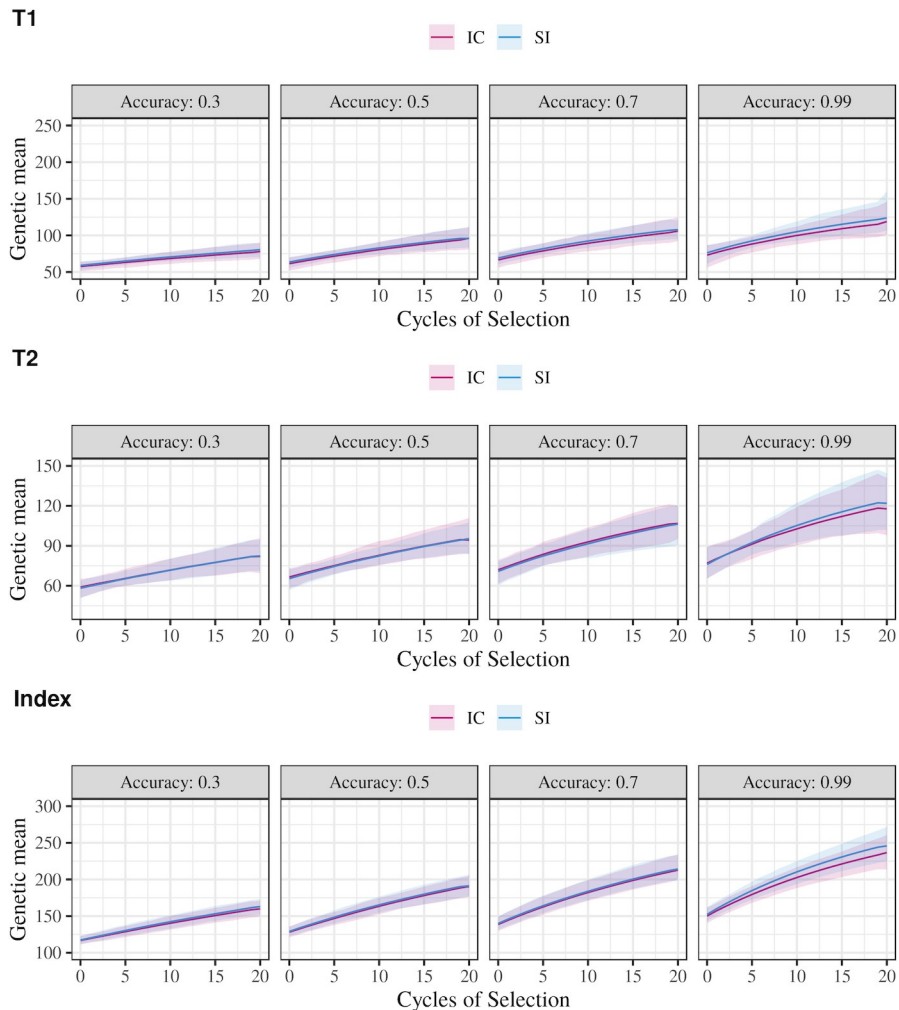

**Fig 3. Change in genetic mean for Trait 1 (T1), Trait 2 (T2) and Index Trait (Index) over 20 cycles of selection using either optimal independent culling (IC) or a selection index (SI) with different levels of accuracy, unfavourably correlated traits, and T2 relative economic importance of 1.0.**

Table 3 shows the genic standard deviation of parents in cycle 20 and the loss in genic standard deviation in cycle 20 compared to the genic standard deviation in cycle 0 for the component traits and the index trait. The values of genic standard deviation of T1, T2, and the index trait were similar. The highest difference between methods in the loss in genic standard deviation was 1% for all values of accuracy, except with accuracy of 0.99. With 0.99 accuracy, for T1, T2 and the index trait, index selection led to a loss in the genic standard deviation that was 3% higher than the loss of genic standard deviation observed using independent culling.

## Relative economic importance of traits

Fig 4 shows the efficiency of converting genetic diversity into genetic gain for the index trait when the relative economic importance of T2 varies. Independent culling was compared to index selection using either optimal culling levels or selection with the same proportion of plants selected (10%) for each trait. Index selection had the highest efficiency and most gain for all levels of economic importance. The efficiency and gain for optimal independent culling

**Table 2. Mean genetic standard deviation (Genetic SD) of parents in cycle 20 and loss in genetic standard deviation in cycle 20 in comparison to the genetic standard deviation in cycle 0 (Loss over cycle 0) for trait 1 (T1), trait 2 (T2) and the index trait using either optimal independent culling or index selection with different levels of accuracy, unfavourably correlated traits, and T2 relative economic importance of 1.0.**

| | Independent culling | | | | | |
|---|---|---|---|---|---|---|
| | **T1** | | **T2** | | **Index trait** | |
| **Accuracy** | **Genetic SD (cycle 20)** | **Loss over cycle 0** | **Genetic SD (cycle 20)** | **Loss over cycle 0** | **Genetic SD (cycle 20)** | **Loss over cycle 0** |
| 0.3 | 3.51 (0.08)* | -17% | 3.68 (0.08) | -16% | 3.57 (0.06) | -22% |
| 0.5 | 2.56 (0.06) | -30% | 2.45 (0.04) | -28% | 2.69 (0.05) | -32% |
| 0.7 | 1.65 (0.04) | -42% | 1.64 (0.03) | -37% | 1.88 (0.04) | -45% |
| 0.99 | 0.45 (0.01) | -68% | 0.45 (0.01) | -55% | 0.74 (0.02) | -62% |
| | **Index Selection** | | | | | |
| | **T1** | | **T2** | | **Index trait** | |
| **Accuracy** | **Genetic SD (cycle 20)** | **Loss over cycle 0** | **Genetic SD (cycle 20)** | **Loss over cycle 0** | **Genetic SD (cycle 20)** | **Loss over cycle 0** |
| 0.3 | 3.80 (0.09) | -11% | 4.00 (0.09) | -11% | 3.66 (0.08) | -19% |
| 0.5 | 3.19 (0.08) | -17% | 3.19 (0.07) | -14% | 2.57 (0.06) | -33% |
| 0.7 | 2.69 (0.06) | -16% | 2.60 (0.06) | -18% | 1.86 (0.04) | -41% |
| 0.99 | 1.93 (0.4) | -3% | 1.91 (0.04) | -4% | 0.51 (0.01) | -59% |

* standard errors of the estimates are presented in parenthesis.

levels were nearly equal to those of index selection. The efficiency and gain for selecting the same proportion of plants for both traits were lower than those of index selection for all levels of relative economic importances. Index selection was 10%, 128% and 310% more efficient than independent culling using the same proportion of selected plants for relative economic importance of 1.0, 2.5 and 5.0, respectively.

Fig 4 also shows the proportion of plants selected for T1 under optimal independent culling over the different levels of economic importance for T2. The mean proportion selected for T1 varied only slightly over the cycles of selection. The means were 29%, 93%, and 99% for relative economic importances of 1.0, 2.5, and 5.0, respectively. The variation about those means was

**Table 3. Genic standard deviation (Genic SD) of parents in cycle 20 and loss in genic standard deviation in cycle 20 in comparison to the genic standard deviation in cycle 0 (Loss over cycle 0) for trait 1 (T1), trait 2 (T2) and the index trait using either optimal independent culling or index selection with different levels of accuracy, unfavourably correlated traits, and T2 relative economic importance of 1.0.**

| | Independent culling | | | | | |
|---|---|---|---|---|---|---|
| | **T1** | | **T2** | | **Index trait** | |
| **Accuracy** | **Genic SD (cycle 20)** | **Loss over cycle 0** | **Genic SD (cycle 20)** | **Loss over cycle 0** | **Genic SD (cycle 20)** | **Loss over cycle 0** |
| 0.3 | 3.94 (0.06)* | -15% | 4.11 (0.07) | -15% | 4.04 (0.05) | -16% |
| 0.5 | 3.48 (0.06) | -24% | 3.41 (0.05) | -24% | 3.44 (0.04) | -25% |
| 0.7 | 2.94 (0.04) | -34% | 2.89 (0.04) | -34% | 2.89 (0.04) | -34% |
| 0.99 | 2.35 (0.04) | -42% | 2.35 (0.04) | -42% | 2.33 (0.04) | -43% |
| | **Index Selection** | | | | | |
| | **T1** | | **T2** | | **Index trait** | |
| **Accuracy** | **Genic SD (cycle 20)** | **Loss over cycle 0** | **Genic SD (cycle 20)** | **Loss over cycle 0** | **Genic SD (cycle 20)** | **Loss over cycle 0** |
| 0.3 | 3.92 (0.06) | -16% | 4.08 (0.07) | -16% | 4.02 (0.05) | -16% |
| 0.5 | 3.44 (0.06) | -25% | 3.37 (0.05) | -25% | 3.39 (0.05) | -26% |
| 0.7 | 2.92 (0.05) | -34% | 2.88 (0.05) | -34% | 2.87 (0.04) | -35% |
| 0.99 | 2.21 (0.04) | -45% | 2.22 (0.04) | -45% | 2.17 (0.03) | -46% |

* standard errors of the estimates are presented in parenthesis.

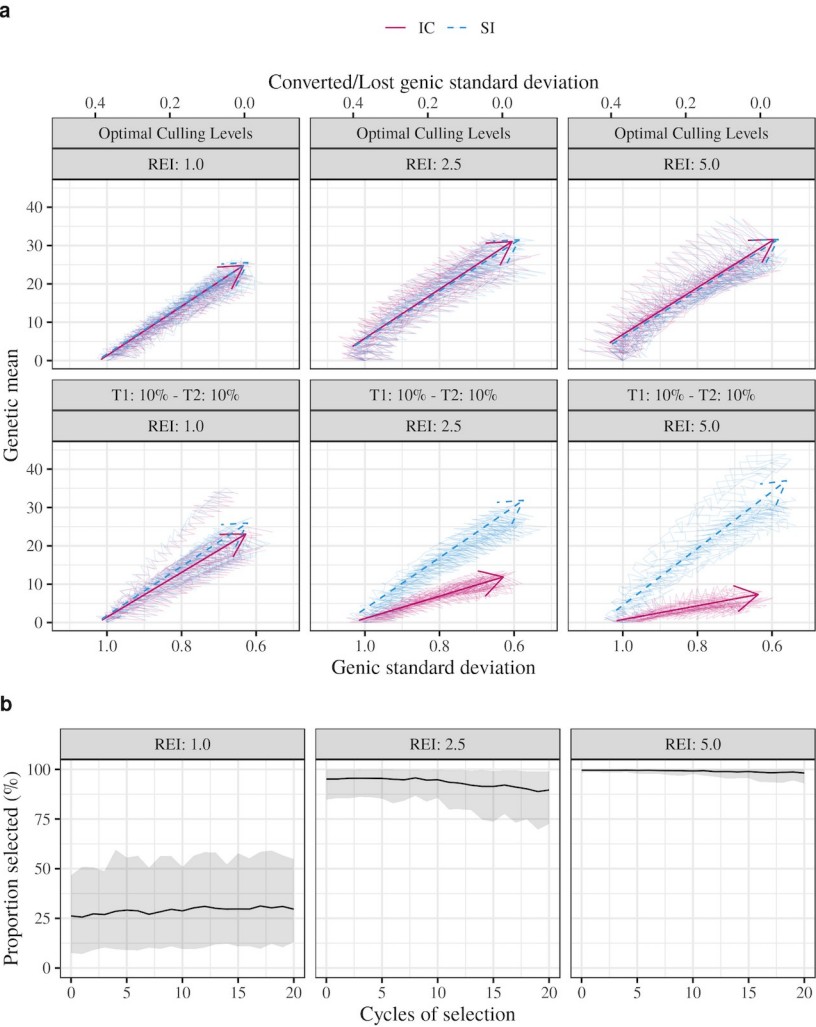

**Fig 4.** Change of genetic mean and genic standard deviation for the index trait across 20 cycles of selection using either independent culling (IC) or a selection index (SI) under three levels of relative economic importance (REI) and using either the same proportion selected (10%) for Trait 1 (T1) and Trait 2 (T2) or optimal culling levels for each level of relative economic importance of T2 (a); and proportion selected (mean and 95% confidence interval) for T1 used to achieve optimal culling levels over the 20 cycles of selection (b). Traits are unfavourably correlated (-0.5). Individual replicates are shown by thin lines and a mean regression with a time-trend arrow. Values of genetic mean and genic standard deviation shown are standardized to mean zero and unit standard deviation in cycle 0.

largest with relative economic importance of 1.0 and smallest with relative economic importance of 5.0.

## Discussion

This study evaluated and compared breeding programs that use either index selection or independent culling for the recurrent selection of parents by genomic prediction. Index selection was either better than or equivalent to independent culling in this context. Index selection outperformed independent culling when a sub-optimal culling level was used.

The main difference between index selection and independent culling is that, under index selection, genotypes that are exceptional for one of the traits under selection are more likely to be selected even though their performance for other traits is average. This can be seen in Fig 1,

with the cluster of individuals selected as parents with the index method including individuals that are more contrasting for the two traits under selection than the individuals selected with independent culling. The main implications of this are in the way each method affects the correlation between traits and the genetic diversity over cycles of recurrent selection. We discuss each of these aspects in the following two sections. Lastly, we discuss how the relative economic importance of the traits can affect the relative performance of the methods.

## Methods of selection and genetic correlation between traits

After only a few cycles of selection, index selection generates $F_1$ populations with a more unfavourable genetic correlation between traits than the $F_1$ populations generated by independent culling (Fig 2). An explanation for the faster decrease of the genetic correlation observed with index selection is that the index is a linear combination of component traits. As shown by Bulmer [28], selection on a linear combination leads to negative covariances between components (the Bulmer effect). Consequently, the same principle applies to the component traits and index selection, with index selection leading to an unfavourable genetic correlation between the component traits [29, 30].

In general, genetic gains in multi-trait selection, regardless of the method of selection, are expected to be higher when the correlation between traits is favourable and lower when this correlation is unfavourable [9]. As index selection generated $F_1$ populations with more unfavourable genetic correlation between traits than independent culling, the genetic gains for index selection were potentially lower than for independent culling. Nevertheless, despite index selection being carried out under increasingly unfavourable genetic correlations over the cycles, the genetic gains obtained for the index trait were equivalent to the gains obtained using independent culling (Fig 3).

Over the cycles of selection, both independent culling and index selection resulted in increasingly unfavourable genetic correlations between traits. Generally, it is assumed that unfavourable genetic correlations that cannot be broken after repeated cycles of recombination are likely due to pleiotropy. This is assumed to be the case in several crops, e.g., grain yield and protein content in cereal crops [31–33], quality and disease resistance in forage crops [34], and yield and disease resistance in barley [35]. However, the extent of the genetic correlation due to pleiotropy in these examples is unknown because, as our study demonstrates, unfavourable genetic correlations between the traits could also be, at least partly, induced by selection.

## Methods of selection and genetic diversity over cycles of selection

According to Bulmer [28], reduction in the genetic variance due to selection stems mostly from the build-up of negative linkage disequilibrium between causal loci when selection is performed. This can be seen by comparing genetic and genic variation (Tables 2 and 3, respectively). Genic variation is a function of the allele frequencies and the allele substitution effect only, and thus is not affected by changes in linkage disequilibrium. The results in Table 3 show that the losses of genic standard deviation of the component traits and index trait were not greatly affected by the method of selection. Also, the method of selection did not greatly affect the trait means, as shown in Fig 3. This indicates that, in terms of allele frequencies, there was little difference in the parents selected by either independent culling or selection index in situations similar to our simulation. Therefore, the difference between the selection methods derives from how they induce and exploit linkage disequilibrium between the causal variants of the component traits. Specifically, as shown in Table 2, independent culling induced a greater degree of negative linkage disequilibrium between the causal variants of the component traits resulting in those traits having less genetic variation. A deviation from this result is

expected with more intense selection schemes and more component traits selected in successive stages, which would induce larger changes in allele frequencies due to drift. As a consequence, differences between index selection and independent culling would be accentuated. In a previous study [36], the authors simulated and compared wheat breeding programs using different selection strategies under high and low selection intensities. They observed that index selection resulted in higher population coancestry over cycles of selection than independent culling, and the difference between methods increased in scenarios with high selection intensity. Their results indicate that index selection leads to a higher loss of genic standard deviation.

Somewhat surprisingly, it is possible to make an argument for the superiority of independent culling relative to a selection index on the basis of the differences observed in linkage disequilibrium. This is because independent culling produced populations with nearly equal mean performance, but with more consistent performance between individuals, as demonstrated by the lower variation observed for the component traits. This property could be beneficial from a management perspective if differences in the component traits require variations in management of individuals. Breeding for plant-architecture traits in outbreeding cultivars is a good example where this property might be valuable, as having more uniform plants in the field favours mechanical harvest. However, we believe this property is more of an academic curiosity than something that will have practical application.

For simplicity and ease of implementation, our simulations considered the same genetic architecture for both traits, with both traits being controlled by a high number (10,000) of causal loci with small additive effects. Under different circumstances, such as at least one of the traits being controlled by few causal loci with higher allele substitution effects, different results could be expected. The results for the two-locus model in [37] show that independent culling tends to eliminate genotypes that are homozygous for alleles with low effect for one of the traits. For one pleiotropic causal locus, when both alleles are favourable for one trait and unfavourable for the other trait, both homozygous genotypes tend to be culled, and independent culling would select the heterozygous genotypes. If heterozygous genotypes were preferred, the fixation of alleles would be slower and, therefore, the loss in genic standard deviation would be lower. Our results indicate that, for highly polygenic traits, differences between methods of selection in the loss of genetic diversity are mostly due to changes in linkage disequilibrium as opposed to distinct changes in allele frequencies. Therefore, in terms of conserving genetic diversity there was no obvious advantage for either method. Other strategies such as optimal-cross selection [26, 36, 38, 39] or the multi-objective optimized approach [21] should be considered in order to optimize gains while also controlling the loss of genetic diversity over cycles of selection.

### Economic importance of the traits

In general, when the same selection intensity is applied to both traits, index selection will perform better than independent culling as the difference in the economic importance of the traits increases (Fig 4). Optimal independent culling performed better than independent culling using the same selection intensity for both traits for all levels of relative economic importance, including the scenario where traits had the same economic importance. The results in Fig 4 show that, when traits had the same economic importance, independent culling approached its maximal gain when a higher selection intensity was used for T1 and a lower selection intensity was used for T2. This indicates that the economic importance of the traits is not the only factor affecting the estimates of optimal culling levels and that accurately estimating them is essential for maximizing gains when independent culling is performed. The results also show that the effect of the economic importance of the traits in the estimates of optimal culling levels

becomes more pronounced with increasing differences in the economic importance of the traits. In fact, when one trait had 5 times the economic importance of the other trait, the optimum was achieved when almost no selection was applied for the less important trait.

Our results show that the two ways of incorporating the true economic weights of the traits in the selection process, either by optimal culling levels or a selection index, lead to nearly equal genetic gains. However, it is worth noting that optimal independent culling would require the complex estimation of optimal culling levels for each trait [11, 14, 15]. When parents are selected based on an index, optimal gain is achieved by simply summing the values of the traits weighted by their economic importance, a much simpler way of maximizing genetic gains in a breeding program.

There is little to no evidence suggesting that plant breeders use analytical techniques to determine optimal independent culling thresholds and/or constructing selection indices in most plant breeding programs. More likely, the majority of breeders rely on their intuition for setting thresholds and constructing indices. Their decisions are likely guided by the performance of agronomic checks and are prone to fluctuations between seasons and individual breeders. This model has clearly been successful, because plant breeding programs have continued to deliver genetic gain. However, it is likely sub-optimal, and the value of a more analytical approach becomes greater as genomic selection is more widely used.

## Conclusions

We evaluated and compared breeding programs using either independent culling or index selection for recurrent parent selection with genomic prediction. Even in the presence of unfavourable genetic correlations, index selection achieved genetic gains equal to or greater than those achieved with independent culling in all simulated scenarios. In terms of genetic diversity, the differences between methods in the studied system were driven mostly by differences in the generation of linkage disequilibrium between causal loci induced and not by differences in allele frequencies. When linkage disequilibrium was not considered, the two methods had similar losses of genetic diversity, and the differences in efficiency of converting genetic diversity into genetic gains between the methods mostly reflected the differences in the genetic gains obtained with each method. To obtain higher genetic gains, accurately estimating optimal culling levels is essential for maximizing gains when independent culling is performed. Once optimal culling levels are estimated, independent culling and index selection lead to nearly equal genetic gains.

## Supporting information

**S1 File.**
(DOCX)

**S2 File.**
(PDF)

## Author Contributions

**Conceptualization:** Peter Amer, John M. Hickey.

**Formal analysis:** Lorena G. Batista.

**Investigation:** Lorena G. Batista.

**Methodology:** Lorena G. Batista.

**Software:** Robert Chris Gaynor.

**Supervision:** Robert Chris Gaynor, Gabriel R. A. Margarido, Tim Byrne, Peter Amer, Gregor Gorjanc, John M. Hickey.

**Validation:** Lorena G. Batista.

**Visualization:** Lorena G. Batista.

**Writing – original draft:** Lorena G. Batista.

**Writing – review & editing:** Robert Chris Gaynor, Gabriel R. A. Margarido, Tim Byrne, Peter Amer, Gregor Gorjanc, John M. Hickey.

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
