## [Decision Letter · Decision Letter 0]

2 Apr 2020

PONE-D-20-04603

An economic selection index should be used instead of independent culling in plant breeding programs with genomic selection

PLOS ONE

Dear Dr. Batista,

Thank you for submitting your manuscript to PLOS ONE. After careful consideration, we feel that it has merit but does not fully meet PLOS ONE’s publication criteria as it currently stands. Therefore, we invite you to submit a revised version of the manuscript that addresses the points raised during the review process.

We would appreciate receiving your revised manuscript by May 17 2020 11:59PM. To enhance the reproducibility of your results, we recommend that if applicable you deposit your laboratory protocols in protocols.io, where a protocol can be assigned its own identifier (DOI) such that it can be cited independently in the future. For instructions see: http://journals.plos.org/plosone/s/submission-guidelines#loc-laboratory-protocols

We look forward to receiving your revised manuscript.

Kind regards,

Roberto Fritsche-Neto, Ph.D.

Academic Editor

PLOS ONE

We note that one or more of the authors are employed by a commercial company: Abacusbio.

"LGB was supported by the Coordenação de Aperfeiçoamento de Pessoal de Nível Superior (CAPES, Computational Biology Programme, Grant No. BEX 0043/17-6). JMH, RCG and GG acknowledge the financial support from BBSRC and KWS UK, RAGT Seeds Ltd., Elsoms Wheat Ltd and Limagrain UK for the project “GplusE: Genomic selection and Environment modelling for next generation wheat breeding” (grants BB/L022141/1 and BB/L020467/1)."

Reviewers' comments:

Reviewer's Responses to Questions

**Comments to the Author**

1. Is the manuscript technically sound, and do the data support the conclusions?

Reviewer #1: Partly

Reviewer #2: Yes

2. Has the statistical analysis been performed appropriately and rigorously? 

Reviewer #1: Yes

Reviewer #2: Yes

3. Have the authors made all data underlying the findings in their manuscript fully available?

Reviewer #1: Yes

Reviewer #2: No

4. Is the manuscript presented in an intelligible fashion and written in standard English?

Reviewer #1: Yes

Reviewer #2: Yes

5. Review Comments to the Author

Reviewer #1: The study idea is good. However, there are some points in the text that should be better described or explained. They are all highlighted in the text. My main objection to the study is to have considered/simulated a very specific selection condition for such a broad objective. In my opinion, the simulated condition responds only to one condition, which involves negatively correlated traits. The correlation between traits is not always negative, nor is it caused by pleiotropy. Thus, it cannot be extrapolated to the whole as the study intends. I suggest that the title, objective and discussion be adjusted to a more restricted selection situation and not as wide as the study contemplated.

Reviewer #2: Review for: An economic selection index should be used instead of independent culling in plant breeding programs with genomic selection

This paper compares two different selection strategies, namely, independent culling and index selection in the context of multi-trait genomic selection. Simulations have been performed for two negatively correlated traits over 20 cycles of breeding for 9 different scenarios with varying selection accuracy and relative economic importance. Results demonstrate, given the economic importance of each trait, maximum genetic gains are more easily achieved with index selection. This study does not provide any new approach or methodology in terms of multi-trait selection and assumes the accurate knowledge of economic importance of traits. However, the reviewer believes it can be useful to the practitioners of genomic selection.

The manuscript is generally well written, and the discussions seem sound. There are some typos throughout the paper which would benefit from proofreading. The figures are not very clear, the authors should provide figures with a higher resolution.

Here are a few specific comments, suggestions and discussion points:

L93-94: What if we select for both traits simultaneously?

L142: It would be worthwhile to consider nonlinear selection indices as well.

L143-147: How are the selected parents mated? Does AlphaSimR have any strategies for mating?

L153: Report the correlation value between two traits.

L155, Table 1: What are the optimum values? These values should be reported. Figure 4 has demonstrated the proportion selected for different relative economic importance, but it would be better to report these variables earlier when showing the results in Table 1.

L203, Figure 1: This Figure demonstrates the genetic values for the selected parents and the F1 population for both traits in the 3rd cycle. It will be interesting to see the same Figure for the final generation (the 20th cycle).

L220, Figure 2: For 0.99 accuracy, the correlation has decreased in the first cycle when using independent culling. How do you describe/interpret that?

L299: The proportion of selected parents for REI=5 is 99%. I wonder what happens if we have a large REI, say 20.

Data availability: Is the simulated data available? I couldn’t find any source links to that.

6. PLOS authors have the option to publish the peer review history of their article (what does this mean?). If published, this will include your full peer review and any attached files.

Reviewer #1: No

Reviewer #2: No

---

## [Author Response · Author response to Decision Letter 0]

16 May 2020

Thank you for these comments. We feel that they have improved the manuscript. 

We have dealt with each of the comments. In what follows we describe our response and give a line number to each change made in the manuscript. For ease of review these changes are also highlighted in yellow in the resubmitted manuscript.

Reviewer #1: The study idea is good. However, there are some points in the text that should be better described or explained. They are all highlighted in the text. My main objection to the study is to have considered/simulated a very specific selection condition for such a broad objective. In my opinion, the simulated condition responds only to one condition, which involves negatively correlated traits. The correlation between traits is not always negative, nor is it caused by pleiotropy. Thus, it cannot be extrapolated to the whole as the study intends. I suggest that the title, objective and discussion be adjusted to a more restricted selection situation and not as wide as the study contemplated.

Response: The reviewer has a fair point. In the beginning of the development of the study we also simulated scenarios with favourably correlated traits. However, the results showed the same patterns observed for unfavourably correlated traits, only less pronounced, as there were more coincidences in the individuals being selected by the two methods (which is expected when traits are favourably correlated). We chose to keep the results that made more evident the differences between methods. Besides that, unfavourable genetic correlations are ubiquitous in breeding programs and are the main challenge faced by breeders deploying multi-trait selection. Both in our introduction (L36) and discussion (L385-387) we provide evidence of unfavourable genetic correlations between traits occurring in several economically important crops.

Given that the scenarios investigated in this study are relevant to the great majority of breeding programs, we consider our title to be appropriate even if we do not show results for other specific scenarios. We have however added clarifications to the introduction (L93) and to the discussion (L379-382) and also added a figure with results from a scenario with favourably correlated traits to our supplementary material.

L55 - Reviewer comment:

“But in traditional breeding, individuals without information are penalized and may not be selected, correct?

Is your claim in the context of genomic selection? I suggest detailing more!”

Response: This claim was made in the context of phenotypic selection. The purpose of the sentence is to compare independent culling to index selection. Index selection requires phenotypes for all traits to be available, whereas with independent culling selection can be made one trait at a time. To make this distinction clearer, we have removed “for all individuals” from the sentence (L55)

L67: Reviewer comment (highlighted the word “render”):“to better?”

Response: We have changed the word “render” to “result in” (L67)

L110-112: Reviewer comment: “Measures with based on any study?”

Response: The reviewer has a fair point. 

These are AlphaSimR default values for the species “wheat”. Given that our goal was to simulate a hypothetical crop species, we considered realistic to use values already observed for a commercial crop such as wheat. We made that explicit and also included the reference: 

L111-118

L119-120: Reviewer comment

“It was not clear to me why the traits were treated as pleiotropic.

This may have generated negative correlations between traits over the generations. But it is not a situation that can be generalized.

Needs explanations!”

Response: The reviewer has a valid point. AlphaSimR only simulates pleiotropic traits, but we believe our results can still be generalized for quantitative traits. 

In our simulations we have 10,000 causal loci, each with a specific additive allele effect for each simulated trait. These allele effects are randomly assigned, resulting in loci having many possible combinations of additive effects for each trait. The loci can be assigned additive effects with the same sign (positive or negative) for both traits, they can be assigned additive effects with opposite signs for the traits, and they can be assigned additive effects with varying absolute values for each trait (i.e. a high effect for one trait and a low effect for the other trait). These are all realistic assumptions under the infinitesimal model, where both traits are controlled by an infinite number of loci with different additive effects for each trait. It also means that although QTL are pleiotropic, they are implicitly acting as non-pleiotropic (i.e., genetic correlations will also be generated by negative linkage disequilibrium). Mechanistically this is the only way to our knowledge to simulate genetic correlations in a controlled way. 

L143: Reviewer comment

“Were these F1s obtained from the 50 founder genotypes? To explain better! To account for this number of descendants, heterozygous parents were probably considered”

Response: The founder genotypes served as the initial parents in the burn-in phase and generated the first F1 population. Genotypes were then selected from the F1 population to be used as parents in the subsequent cycle. We have now modified the sentence to make this clearer.

L143-146

L154: Reviewer comment

“It was not clear to me why this correlation was considered negative. The simulation refers to a specific selection situation and not what occurs in general.”

Response: This is described in lines 165-167. Our intention was to simulate a challenging scenario for the breeder. Unfavourable genetic correlations can be either positive or negative, depending on the direction of the selection being carried for each trait. In our simulations we were selecting for higher genetic values for both traits, hence we used a negative genetic correlation as unfavourable. A positive genetic correlation would be unfavourable if the direction of selection was different between traits.

Moreover, for favourably correlated traits, both methods perform very similarly, because most of the individuals selected by index selection would also be selected by independent culling. We considered that scenarios with favourable genetic correlation would not be of much additional value to the manuscript and we decided to report and discuss only results from simulations which represent challenging scenarios for multi trait selection, and which would differentiate the two methods of selection being evaluated. We have added clarifications to the introduction (L93), results (L219-222 and 228-229) and to the discussion (L379-382) and also added a figure with results from a scenario with favourably correlated traits to our supplementary material.

L167-168: Reviewer comment:

“This has a consequence for genetic diversity. This does not reflect reality”

Response: The reviewer is right. Our goal in the paper was to compare index selection to the best possible results that can be obtained with independent culling, hence the optimal was chosen. Nevertheless, we also report and discuss the results (Fig. 4) for independent culling when a suboptimal proportion selected was used for each trait. The results are discussed in terms of efficiency in converting genetic diversity into genetic gain.

L240-243: Reviewer comment

“Probably linked to the maximization of genetic gains that promoted a more drastic reduction”

Response: As stated in lines 407-409 of our discussion, after comparing the genetic standard deviation with the genic standard deviation, we believe this difference between the selection methods stem from the how these methods are inducing and exploiting linkage disequilibrium between the causal variants.

L336-339: Reviewer comment

“It is a very specific situation. What if the correlation was positive? I would like to see this situation.”

Response: For favourably correlated traits, both methods would perform very similarly, because most of the individuals selected by index selection would also be selected by independent culling.

We have however added clarifications to the discussion (L379-382) and also added a figure with results from a scenario with favourably correlated traits to our supplementary material.

L394-396: Reviewer comment

“I agree. However, in practice I do not know of a breeding program for clonally propagated species conducted over many cycles. I believe it is impractical.”

Response: The reviewer has a point. Conventional breeding programs for clonally propagated crops take several years for development of cultivars, which are generally used as parents in the next breeding cycle. 

As shown by Gaynor et al. (2017), genomic selection allows breeding programs to be restructured by decoupling their cultivar development component from their population improvement component. As the population improvement component operates independently, this enables rapid recurrent selection with potentially several cycles of selection per year to be carried. In this context, the loss of genetic diversity needs to be carefully managed. 

L446-447: Reviewer comment

“economics weights of the traits?”

Response: The reviewer is correct. We have modified the sentence:

L511-512

Reviewer #2: Review for: An economic selection index should be used instead of independent culling in plant breeding programs with genomic selection

This paper compares two different selection strategies, namely, independent culling and index selection in the context of multi-trait genomic selection. Simulations have been performed for two negatively correlated traits over 20 cycles of breeding for 9 different scenarios with varying selection accuracy and relative economic importance. Results demonstrate, given the economic importance of each trait, maximum genetic gains are more easily achieved with index selection. This study does not provide any new approach or methodology in terms of multi-trait selection and assumes the accurate knowledge of economic importance of traits. However, the reviewer believes it can be useful to the practitioners of genomic selection.

The manuscript is generally well written, and the discussions seem sound. There are some typos throughout the paper which would benefit from proofreading. The figures are not very clear, the authors should provide figures with a higher resolution.

Response: We thank the reviewer for the comments. We have proofread the manuscript again and found a number of typos. We apologise for their presence in the earlier version. Regarding the submitted figures, we used the highest resolution allowed in the Journal submission guidelines. 

Here are a few specific comments, suggestions and discussion points:

L93-94: What if we select for both traits simultaneously?

Response: Independent culling consists of selection for each trait being carried independently of the other traits, even if selection for multiple traits is carried simultaneously. The genotypes being selected would have been the same, although cycle time would be shorter.

L142: It would be worthwhile to consider nonlinear selection indices as well.

Response: The reviewer has a good point and our group is planning subsequent research using non-linear indexes as well. In this study, we decided to only consider linear selection indexes. We address this in L489-491 of our manuscript, in the discussion section. 

L143-147: How are the selected parents mated? Does AlphaSimR have any strategies for mating?

Response: In this study the parents were randomly crossed. AlphaSimR also allows user-supplied crossing plans. 

L153: Report the correlation value between two traits.

Response: The reviewer is correct. We have added this information in L167-168. 

L155, Table 1: What are the optimum values? These values should be reported. Figure 4 has demonstrated the proportion selected for different relative economic importance, but it would be better to report these variables earlier when showing the results in Table 1.

Response: Table 1 is not in the results section so for this reason we excluded the results there. The optimal values of proportion selected varied across different replicates of the simulations, which is shown by the confidence margins in Figure 4. We have now provided the mean values of proportion selected for each cycle in Table S1.1 of our supplementary material in addition to the figure.

L203, Figure 1: This Figure demonstrates the genetic values for the selected parents and the F1 population for both traits in the 3rd cycle. It will be interesting to see the same Figure for the final generation (the 20th cycle).

Response: The purpose of this figure is to illustrate which individuals were being selected by each of the methods. The characteristic pattern of individuals being selected by each method does not change over the cycles of selection. We have added the figure for the 20th cycle to the supplementary results and included a sentence summary of the similarities and differences between the 1st and the 20th cycles on lines L219-223. 

L220, Figure 2: For 0.99 accuracy, the correlation has decreased in the first cycle when using independent culling. How do you describe/interpret that?

Response: Very good question. In the burn-in phase we used independent culling as the selection method. Hence, for the independent culling scenarios the first cycle being reported is merely a continuation of the burn-in phase, i.e. there’s nothing specific to the first cycle which differentiates it from the other cycles of selection. Given that the observed change in genetic correlation was not significant, we believe the increase in genetic correlation was not a relevant occurrence. However, we have added a sentence to report this in our results:

Line 238-239

L299: The proportion of selected parents for REI=5 is 99%. I wonder what happens if we have a large REI, say 20.

Response: In this case, results reported in Hazel and Lush (1942) indicate that no selection at all would be carried for the trait with lower economic importance. 

Data availability: Is the simulated data available? I couldn’t find any source links to that.

Response: The reviewer has a valid point. To address this, we decided to make the code we used for the simulations available as supplementary material.

---

## [Decision Letter · Decision Letter 1]

5 Nov 2020

PONE-D-20-04603R1

An economic selection index should be used instead of independent culling in plant breeding programs with genomic selection

PLOS ONE

Dear Dr. Batista,

Thank you for submitting your manuscript to PLOS ONE. After careful consideration, we feel that it has merit but does not fully meet PLOS ONE’s publication criteria as it currently stands. Therefore, we invite you to submit a revised version of the manuscript that addresses the points raised during the review process.

As you can see, both of the original reviewers were satisfied with your responses to their points. Your manuscript has been passed to me to handle, and I request some clarification and changes.

The main point made in this paper appears to be that independent culling (IC) faces the problem of determination of selection intensity according to the economic weighting of the traits under selection. However, for maximum efficiency of genetic gain, index selection (IS) also requires estimation of these economic weights. I have read and reread the MS and cannot find any admission that when these weights are accurately estimated, the only differences between IC and IS are that IC can be applied sequentially according to when traits are scorable, and that IS allows selection of genotypes with sub-threshold scores for some traits when the remaining traits have superior scores. The authors cite both advantages as having been well known before the study was undertaken.

On line 143 appears “The index trait was the sum of the estimated breeding values for each trait weighted by their economic importance.” It would be helpful to see the mathematical expression for this weighting.

But if we can develop an economic selection index based on known economic weights, why could we not perform IC based on the same weights? The authors say on line 195 “Index selection performed better than independent culling in scenarios where independent culling levels were suboptimal.” Well, sure. You could just as well say that IC performed better than IS in scenarios where IS economic weights were suboptimal. But if you know how to weight one method, you know how to weight the other.

I ask the authors to address this difficulty.

There are many writing and some organizational errors in the text; many of them are listed below with corrections.. The line numbers are those of the authors’ revised submission, PONE-D-20-04603_R1.pdf. The authors are not required to adopt the exact substitute wording shown with each item, but it is strongly recommended.

23 accordingly => according

24 efficiency => efficiencies

25 both => the two

26 proportion selected => selection proportion

27 a relative economic importance => relative economic importances

29 fact => finding

30 to index => to those from index

40 on => for

41 only selecting => selecting only

44 on => to [twice]

46 non-linear => nonlinear

52 equivalent => equal

61 only be measured => be measured only

63 equivalent => equal

69  allows for accurate => allows accurate

76 discussed => shown [studies cannot discuss]

77  In addition, other  => Other

86 quantify the magnitude of the difference => quantify the difference

124  the simulated traits => each simulated trait

128 Where => where

139 Parents => parents

139 carried out => applied

144 [You mention economic importance here, but you have not described how economic importance was assigned.]

153 only interested in investigating => interested in investigating only

154 multi trait => multi-trait [correct this error throughout the MS]

155 Hence => For this reason

163 to => at

163 Here, three => Three

169 the proportion selected for each trait => for each trait the selection proportion [in most cases in the MS you can replace “proportion selected” with “selection proportion”]

174 (i.e. index trait) => (measured as the index trait)

193 Overall the results show that index => Index

201 The results show that increases => Increases

214 was higher with higher values of => increased with

217 compared to => than [please make the same correction throughout, when a comparative expression is used. A few other examples: 221, 256, 271]

243, 245, 314 when using => under

252  loss of genetic standard deviation that was =>  losses of genetic standard deviation that were respectively

253, 257 , respectively. => . [delete this word and preceding comma]

267 The values of genic standard deviation of T1, T2, and the index trait were equivalent. [Equivalent means having the same value. We don’t say that values have the same value. If you are using equivalent to mean similar, please don’t. Just write similar. If you are using it to mean equal, please write equal.]

286 equivalent to index => equal to that of

287 was worse than => were lower than those of

300 plant => plants

301 only varied => varied only

308 either use => use either

310 Overall the results show that using index selection is either better => Index selection was either better than

312 Our results demonstrate that accurately...is essential => Accurately...was essential

327 The results show that, after...generates => After...generated

332 (i.e. Bulmer => (the Bulmer

343 - 348 This material is not discussion. It is introduction and belongs only there, as part of the motivation for the study. In Discussion we discuss only the results of the present study and ideas that were first suggested by the results.

357 are => were

370 observed => observed that

372 indicate => indicate that

377 which is => as

384 consider = > considered

395 distinctive => distinct

401  using the same selection intensity for =>  the same selection intensity is applied to

404 when performing => under

412 carried out => applied

412 These results demonstrate that => Thus,

418 using these weights => these weights are used

421 suggesting => suggesting that

422 constructing => construct

430 show => show that

431 when using => in

433 Miscanthus => [italicize this genus name]

433 This [what is “this”?]

435 be => be on

436 paper => study

437 the the => the

437 and it is => and is

438 non-linear => nonlinear [two places]

443  that suggest selection weights should be => in which selection weights are

445 eigen selection => [Eigen is not a word by itself. Consider eigenselection or eigenvalue selection, whatever the original author used]

445 do => does

447 On => Under

447 carried => made

455  The results show that, despite selection being carried out under unfavourable genetic correlations when using the selection index instead of independent culling, equivalent or higher genetic gains were achieved with index selection in all simulated scenarios => Even in the presence of unfavourable genetic correlations, index selection achieved genetic gains equal to or greater than those achieved with independent culling.

460 not => not by

461 both => the two [it is meaningless to say that one method was equivalent and so was the other]

We look forward to receiving your revised manuscript.

Kind regards,

James C. Nelson, Ph.D.

Academic Editor

PLOS ONE

Reviewers' comments:

Reviewer's Responses to Questions

**Comments to the Author**

1. If the authors have adequately addressed your comments raised in a previous round of review and you feel that this manuscript is now acceptable for publication, you may indicate that here to bypass the “Comments to the Author” section, enter your conflict of interest statement in the “Confidential to Editor” section, and submit your "Accept" recommendation.

Reviewer #1: All comments have been addressed

Reviewer #2: All comments have been addressed

2. Is the manuscript technically sound, and do the data support the conclusions?

Reviewer #1: Yes

Reviewer #2: Yes

3. Has the statistical analysis been performed appropriately and rigorously? 

Reviewer #1: Yes

Reviewer #2: Yes

4. Have the authors made all data underlying the findings in their manuscript fully available?

Reviewer #1: Yes

Reviewer #2: Yes

5. Is the manuscript presented in an intelligible fashion and written in standard English?

Reviewer #1: Yes

Reviewer #2: Yes

6. Review Comments to the Author

Reviewer #1: The authors answered or clarified all doubts and suggestions and, therefore, I recommend the publication of the paper.

Reviewer #2: The authors have addressed all comments raised in the last round. The reviewer does not have additional comments.

7. PLOS authors have the option to publish the peer review history of their article (what does this mean?). If published, this will include your full peer review and any attached files.

Reviewer #1: No

Reviewer #2: No

---

## [Author Response · Author response to Decision Letter 1]

2 Dec 2020

Response to Academic Editor

Editor comment:

“As you can see, both of the original reviewers were satisfied with your responses to their points. Your manuscript has been passed to me to handle, and I request some clarification and changes.”

Response: We’d like to thank the Editor for taking over the review process of the manuscript. We appreciate that the review process is now moving forward in the fairest way possible. 

Throughout this rebuttal letter we will use the line numbers in the “Manuscript.docx” file. Line numbers are indicated by the letter “L” followed by the line number. 

Editor comment:

“The main point made in this paper appears to be that independent culling (IC) faces the problem of determination of selection intensity according to the economic weighting of the traits under selection. However, for maximum efficiency of genetic gain, index selection (IS) also requires estimation of these economic weights. I have read and reread the MS and cannot find any admission that when these weights are accurately estimated, the only differences between IC and IS are that IC can be applied sequentially according to when traits are scorable, and that IS allows selection of genotypes with sub-threshold scores for some traits when the remaining traits have superior scores. The authors cite both advantages as having been well known before the study was undertaken.

On line 143 appears “The index trait was the sum of the estimated breeding values for each trait weighted by their economic importance.” It would be helpful to see the mathematical expression for this weighting.”

Response: We agree with the Editor and have included the formula in matrix notation in the manuscript (L142 – L146)

Editor comment:

“But if we can develop an economic selection index based on known economic weights, why could we not perform IC based on the same weights? The authors say on line 195 “Index selection performed better than independent culling in scenarios where independent culling levels were suboptimal.” Well, sure. You could just as well say that IC performed better than IS in scenarios where IS economic weights were suboptimal. But if you know how to weight one method, you know how to weight the other. I ask the authors to address this difficulty.”

Response: The Editor has a valid point and we modified the discussion in order to address his comments. 

L414 – L419: we make more explicit in our discussion that both independent culling and index selection could lead to optimal genetic gain. However, we also considered important to emphasize that finding optimal culling levels is a much more complex task than using an index (see references 11,38,39,40,41,42 and 43). Therefore, index selection should be preferred. 

Editor comment:

“There are many writing and some organizational errors in the text; many of them are listed below with corrections. The line numbers are those of the authors’ revised submission, PONE-D-20-04603_R1.pdf. The authors are not required to adopt the exact substitute wording shown with each item, but it is strongly recommended.”

Response:

We thank the Editor for his thorough review and for pointing out the many errors we have overlooked. We have modified the manuscript in accordance with all of the suggestions. Some of the suggestions are presented below followed by a response. 

"144 [You mention economic importance here, but you have not described how economic importance was assigned.]"

Response: The description requested can be found in L166 – L168.

"343 - 348 This material is not discussion. It is introduction and belongs only there, as part of the motivation for the study. In Discussion we discuss only the results of the present study and ideas that were first suggested by the results."

Response: The Editor has a valid point and we modified the paragraph in order to keep only what is relevant for the discussion of the results (L344 – L351). 

"433 This [what is “this”?]"

Response: The sentences were modified in order to be more specific (L433)

---

## [Editor Report · Decision Letter 2]

15 Dec 2020

PONE-D-20-04603R2

An economic selection index should be used instead of independent culling in plant breeding programs with genomic selection

PLOS ONE

Dear Dr. Batista,

Thank you for submitting your manuscript to PLOS ONE. After careful consideration, we feel that it has merit but does not fully meet PLOS ONE’s publication criteria as it currently stands. Therefore, we invite you to submit a revised version of the manuscript that addresses the points raised during the review process.

We look forward to receiving your revised manuscript.

Kind regards,

James C. Nelson, Ph.D.

Academic Editor

PLOS ONE

Additional Editor Comments (if provided):

Rather than ask further reviewers to read a submission I find confusing, I have chosen to try to clarify it myself. I am sorry to keep pointing out things I consider problematic, but every time I read the revised MS I find more problems and less substance. I hope these remarks will help the authors in communicating what they did and in separating what this study really established from what they could have written without ever having done a study. I can’t yet distinguish a failure to communicate from a failure of experimental rigor, but both are fatal. PLoS One standards will not prevent publication of a study of light weight, so long as it was done with rigor, and I would add: that readers can tell what it actually contributes to the field, however little that may be.

In short: what did you find out that you didn’t already know, and couldn’t have predicted from prior knowledge? That most of your Discussion is just introduction and is independent of your experimental results is troublesome. I ask the authors to define “accuracy” as computed in their simulation study, and to remove from Discussion all material that was not first suggested by the results (details below). I ask them to clarify how selection was performed under independent culling (details below). I ask them to acknowledge, if they agree, that all they have shown about the non-optimality of independent culling is that a flat 10% selection intensity is not as effective (by their chosen criteria) as index selection.

164 and Table 1 Four levels of accuracy... => You have nowhere defined accuracy or explained how these levels were applied to the simulation. You could presumably show this mathematically or, failing that, algorithmically. Even up to line 201, citing “the accuracy of selection”, you have not defined accuracy. The paper simply makes no sense without such a definition.

171-172 Maximised the genetic gain [to establish culling levels]? The true genetic gain is not known to a breeder. If you are using that, the simulation is not realistic. This is important, because your conclusions cite “optimal” culling levels. Did you maximise the genetic gain estimated from the phenotype, a parameter that would be available to a breeder? But then in line 176-177 you write that you did not use genetic gain at all, but economic value, and applied index selection to determine culling levels. Supplementary figure S.1.1 shows that this paragraph mixes the two methods without distinction. But the main MS should not depend on the supplementary material for explanation.

198 independent culling levels were suboptimal => This is deceptive, because you did not simulate “suboptimal levels”. You simulated only one, a 10% selection proportion.

313 Accurately assessing the economic importance of the traits is essential => Where is the experimental evidence for this assertion, given that you have not defined accuracy or, as far as your M&M indicates, simulated any process by which a breeder might assess economic importance? Is the 10% selection variant the only test you are using to represent “inaccurate” estimation of economic importance?

412 Same as 313. How can you make this assertion?

358 loss => losses

402-407, 414-419; this was all known before the study and is not discussion of the results.

412-413. You did not test estimation errors; you compared only the flat 10% (which no sensible breeder would be likely to apply for traits with clear differences in economic importance) with projection of genetic gain.

421-428 This material is not discussion of your results and is not a conclusion from your results. You knew it from the literature before the study was performed. The same is true of 435-441; you could have written it without doing any experiment. Your simulation doesn’t include any test of estimation errors of culling levels.

429-434 All except the second sentence are introduction. They serve as background knowledge and motivation for the decision to perform this study. The second sentence just repeats a claimed finding; repetition doesn’t add evidence.

435-452 is pure introduction. It is not derived from any findings of this study.

463-466 is introduction. That optimal culling levels are complex to estimate was given, not shown in the study.

As a reminder: an Introduction poses the research problem and says what we know about it, what we wish to find out, and how we propose to find it out.

The M&M and Results section seem to be in order in this MS, except for the omission in explaining “accuracy” and how it was simulated.

A Discussion explains and draws inferences from the results. Unlike the Introduction, it’s concerned with presenting ideas that we didn’t already know and that were first suggested by the results. If there is little to discuss, the section should be short. Introductory material, such as composes much of the Discussion in this MS, should be moved to the Introduction, unless it is mere repetition of material that is already there.

---

## [Author Response · Author response to Decision Letter 2]

13 Jan 2021

Editor Comment: 

“Rather than ask further reviewers to read a submission I find confusing, I have chosen to try to clarify it myself. I am sorry to keep pointing out things I consider problematic, but every time I read the revised MS I find more problems and less substance. I hope these remarks will help the authors in communicating what they did and in separating what this study really established from what they could have written without ever having done a study. I can’t yet distinguish a failure to communicate from a failure of experimental rigor, but both are fatal. PLoS One standards will not prevent publication of a study of light weight, so long as it was done with rigor, and I would add: that readers can tell what it actually contributes to the field, however little that may be.”

Response: We appreciate and we are thankful for the provided input. We agree that a failure to communicate is aggravating, hence we modified the manuscript in order to address the issues pointed out in the revision.

Editor Comment: 

“In short: what did you find out that you didn’t already know, and couldn’t have predicted from prior knowledge?” 

Response: Our study has fully detailed the effect of different methods of selection in key population parameters such as genetic diversity, genetic correlation and genetic gain over several cycles of selection. No other study has compared index selection and independent culling for that many cycles of selection. Some of the results we observed were expected and consistent with previous literature, some of the results were somewhat surprising, like the increasingly unfavourable genetic correlation between traits (which the Bulmer effect eventually helped us understand). 

Editor Comment: 

“That most of your Discussion is just introduction and is independent of your experimental results is troublesome.” 

Response: We thank the editor for pointing this out, and we have moved some of the paragraphs from the Discussion to the introduction.

These paragraphs arose as a result from many discussions with other researchers and plant breeders while presenting our results. We wrongly decided to include them as a part of our discussion. However, not with the intent of claiming these findings to our study, but as important takeaway messages to breeders. 

Editor Comment: 

“I ask the authors to define “accuracy” as computed in their simulation study, and to remove from Discussion all material that was not first suggested by the results (details below).I ask them to clarify how selection was performed under independent culling (details below). I ask them to acknowledge, if they agree, that all they have shown about the non-optimality of independent culling is that a flat 10% selection intensity is not as effective (by their chosen criteria) as index selection.

164 and Table 1 Four levels of accuracy... => You have nowhere defined accuracy or explained how these levels were applied to the simulation. You could presumably show this mathematically or, failing that, algorithmically. Even up to line 201, citing “the accuracy of selection”, you have not defined accuracy. The paper simply makes no sense without such a definition.”

Response: We defined accuracy as the correlation between true and estimated breeding values. This definition is present in the manuscript. We now slightly modified the sentence to highlight it better (L138-L139)

Editor Comment: 

“171-172 Maximised the genetic gain [to establish culling levels]? The true genetic gain is not known to a breeder. If you are using that, the simulation is not realistic. This is important, because your conclusions cite “optimal” culling levels. Did you maximise the genetic gain estimated from the phenotype, a parameter that would be available to a breeder? But then in line 176-177 you write that you did not use genetic gain at all, but economic value, and applied index selection to determine culling levels. Supplementary figure S.1.1 shows that this paragraph mixes the two methods without distinction. But the main MS should not depend on the supplementary material for explanation.”

Response: We have included in the manuscript that optimal independent culling aimed at maximizing the genetic gain for the index trait, but it does it by selecting on one trait at a time. We obtained the index trati from estimated breeding values of the traits (L180-L181). We thank the Editor for pointing that out. 

The figure in S.1.1 only represents the selection methods graphically, it doesn’t add information that is not already included in the Material and Methods. 

1) In the independent culling method selection is performed one trait at a time as is often done in plant breeding. Individuals are ranked based on one trait and the individuals with the highest values are kept while the other individuals are culled. The individuals that were kept are then ranked based on a second trait and then culled again. This process is repeated for each additional trait under selection.

2) In the selection index method individuals are selected for all traits at once, generally by using a linear combination of the traits, which is the index. Individuals are ranked based on the index and the ones with the highest values are kept.

Hence, in the optimal independent culling methodology we do not mix the two methods. The index trait is used as a reference for establishing which are the optimal selection intensities we need to use for each trait. Once we have these selection intensities (i.e. the number of individuals we need to cull in each stage) independent culling is carried as described in 1. We did not use the index trait to rank the individuals, selection was carried out for each trait at a time.

Editor Comment: 

“198 independent culling levels were suboptimal => This is deceptive, because you did not simulate “suboptimal levels”. You simulated only one, a 10% selection proportion.”

Response: We agree with the Editor and we have modified the sentence accordingly (L207-208)

Editor Comment: 

“313 Accurately assessing the economic importance of the traits is essential => Where is the experimental evidence for this assertion, given that you have not defined accuracy or, as far as your M&M indicates, simulated any process by which a breeder might assess economic importance? Is the 10% selection variant the only test you are using to represent “inaccurate” estimation of economic importance?

412 Same as 313. How can you make this assertion?”

Response: We agree with the Editor and we have removed the sentences from our manuscript 

Editor Comment: 

“402-407, 414-419; this was all known before the study and is not discussion of the results.

Response: Please see below.

412-413. You did not test estimation errors; you compared only the flat 10% (which no sensible breeder would be likely to apply for traits with clear differences in economic importance) with projection of genetic gain.

Response: Please see below.

421-428 This material is not discussion of your results and is not a conclusion from your results. You knew it from the literature before the study was performed. The same is true of 435-441; you could have written it without doing any experiment. Your simulation doesn’t include any test of estimation errors of culling levels.

Response: Please see below.

429-434 All except the second sentence are introduction. They serve as background knowledge and motivation for the decision to perform this study. The second sentence just repeats a claimed finding; repetition doesn’t add evidence.

Response: Please see below.

435-452 is pure introduction. It is not derived from any findings of this study.

Response: Please see below.

463-466 is introduction. That optimal culling levels are complex to estimate was given, not shown in the study.

As a reminder: an Introduction poses the research problem and says what we know about it, what we wish to find out, and how we propose to find it out.

The M&M and Results section seem to be in order in this MS, except for the omission in explaining “accuracy” and how it was simulated.

A Discussion explains and draws inferences from the results. Unlike the Introduction, it’s concerned with presenting ideas that we didn’t already know and that were first suggested by the results. If there is little to discuss, the section should be short. Introductory material, such as composes much of the Discussion in this MS, should be moved to the Introduction, unless it is mere repetition of material that is already there.”

Response: We agree with the Editor and we have considered all these comments carefully in order to modify the manuscript. Some of the paragraphs in our discussion were excluded from the manuscript and some were moved to the introduction, in accordance with the remarks.

We thank the Editor’s for these comments, and we belie this revision helped us improve our discussion. We also changed our title in accordance to the comments. 

The flat 10% selection intensity for both traits would have been the breeder’s intuitive choice in scenarios with a relative economic importance of 1.0 (traits with equal economic importance). Even in that case, optimal culling levels performed better than the intuitive choice. We have decided to argue in our discussion that this result indicates that culling levels should not be chosen based on intuition and should be estimated accurately aiming at maximizing economic value.

We have kept the remark about the complexity of estimating optimal culling levels in the discussion, but we removed it from our conclusions. Since we showed that both optimal independent culling and index selection lead to equivalent genetic gains, the complexity is an important caveat when deciding between the two methods.

---

## [Editor Report · Decision Letter 3]

25 Jan 2021

Long-term comparison between index selection and optimal independent culling in plant breeding programs with genomic prediction

PONE-D-20-04603R3

Dear Dr. Batista,

We’re pleased to inform you that your manuscript has been judged scientifically suitable for publication and will be formally accepted for publication once it meets all outstanding technical requirements.

Kind regards,

James C. Nelson, Ph.D.

Academic Editor

PLOS ONE

Additional Editor Comments (optional):

Thanks to the authors for their patience — in fact they had explained accuracy and I had overlooked it.

I am recommending acceptance of the MS for publication, but I ask the authors to attend to the corrections and questions below before submitting their final version, which I do not need to see.

Abstract: had equivalent loss => had similar [equal, identical] losses

67 obsolete => irrelevant

77 the correct => accurate

79 for each trait [? Multiple selection intensities for a single trait? Do you mean “for the traits”?]

96 culling a sub-optimal culling level [? Doesn’t make sense. Do you mean “culling using suboptimal culling levels”?]

114 AlphaSimR [This must be defined or cited when first mentioned, not at a later mention. And once it has been cited, it should not be cited again.]

117 1.8x10-9 what? Giraffes?

177 (10%) when selecting for => (10%) for

183-186 Doesn’t this repeat 179-181?

199 mean => [don’t you mean “gain”?]

294-295 was nearly equal to that => were nearly equal to those

260-262 loss ... that was 6% and 5% higher... respectively => losses...that were respectively 6% and 5% higher...

298, 313 importance => importances

320 better or => better than or

383 equivalent => equal

423 both => the two

424 leads => lead

424 by using optimal culling levels or a => by optimal culling levels or by a

428 importance, therefore being a => importance, a

445 equivalent loss => similar losses

446 the efficiency => the differences in efficiency
---

## [Editor Report · Acceptance letter]

24 Jun 2020

PONE-D-20-04603R1 

An economic selection index should be used instead of independent culling in plant breeding programs with genomic selection 

Dear Dr. Batista:

I'm pleased to inform you that your manuscript has been deemed suitable for publication in PLOS ONE. Congratulations! Your manuscript is now with our production department. 

Kind regards, 

on behalf of

Professor Roberto Fritsche-Neto 

Academic Editor

PLOS ONE